# Mediating Effects of the COVID-19 Pandemic on the Associations between Physical Activity and Physical Fitness; Cross-Sectional Study among High School Adolescents

**DOI:** 10.3390/jfmk8030131

**Published:** 2023-09-06

**Authors:** Mirela Sunda, Barbara Gilic, Petra Rajkovic Vuletic, Vladimir Pavlinovic, Natasa Zenic

**Affiliations:** 1Faculty of Kinesiology, University of Zagreb, 10000 Zagreb, Croatia; sundamirela@gmail.com (M.S.); petrarajkovicvuletic@gmail.com (P.R.V.); 2Faculty of Kinesiology, University of Split, 21000 Split, Croatia; barbara.gilic@kifst.eu (B.G.); vladimir.pavlinovic@kifst.eu (V.P.)

**Keywords:** health habits, youth, disease outbreak, clusters, lifestyle medicine

## Abstract

The COVID-19 pandemic and the imposed social distancing measures caused negative changes in physical activity levels (PALs) and physical fitness (PF) among adolescents, but the potential mediating effect of the pandemic on the association between PAL and PF is unknown. This study aimed to evaluate gender-specific associations between objectively measured PAL and indices of PF among high school adolescents during the period of the COVID-19 pandemic. The participants were 150 adolescents (101 females) aged 14–18 years; their anthropometrics were evaluated, and they were tested on PF (cardiorespiratory fitness (CRF; beep test), power (broad jump), flexibility (sit-and-reach test), and abdominal strength (sit-ups)) and PAL (using a pedometer for 7 days) during the pandemic period. A *T*-test was calculated to determine differences between genders. Associations between variables were evaluated using Pearson correlations. Additionally, multivariate taxonomic classification was used to sort participants into homogenous groups (clusters) according to their PF, and then analysis of variance (ANOVA) was performed to differentiate them according to PAL. For the total sample, PAL was correlated with CRF only (R = 0.25, *p* < 0.05), while gender-stratified correlations showed that PAL was significantly associated with CRF among girls only (R = 0.29, *p* < 0.05), which was additionally confirmed with multivariate cluster analysis and subsequent ANOVA. No association between PAL and PF was found for boys. The relatively low association between PAL and PF is most likely related to the mediating effect of the change in life circumstances during the COVID-19 pandemic, and limited applicability of pedometers in evaluating high-intensity PAL. Further studies in other age groups and environments are warranted.

## 1. Introduction

The COVID-19 pandemic that was globally declared at the beginning of 2020 caused enormous changes in everyday life [1]. Imposed social distancing measures and lockdowns resulted in significant decreases in physical activity levels (PALs), especially among children and adolescents, mainly as a result of the ban on organized sports and recreational activities and lack of physical education (since schools were organized using different online models) [2]. Moreover, other factors have also been shown to have an influence on changes in PALs during the pandemic [3]: age (a larger decline in PALs was observed in younger adolescents) [4], gender (a larger decrease in PALs was observed in boys than girls) [5], living environment (a larger decrease in PALs in urban than rural adolescents) [6], and familial/parental factors (adolescents who had conflicts with their parents had lower PALs) [7]. Supporting this, studies confirmed a decrease in PALs during the pandemic period [6,8]. Meanwhile, achieving appropriate PALs in childhood and adolescence is of unquestionable importance, since it is directly related to physical fitness and health status, while physical activity is known to be one of the pillars of lifestyle medicine [9,10].

Adolescence is the life period during which the most significant changes in PALs occur [11]. Dropping out from sports, having decreased free play, and having increased screen time are known to be the most important determinants of such changes [12]. Indeed, reports are consistent in concluding that PALs drop the most during the transition from childhood to adolescence [13,14]. In fact, a global report on physical activity among adolescents revealed alarming figures: only 19% of adolescents worldwide reach the recommended PAL, a minimum of 60 min of moderate to vigorous physical activity a day [15]. The same trend was found among Croatian adolescents, with only 23% of adolescents (21% of girls, 27% of boys) reaching the physical activity standards [16]. This is worrying, considering that an appropriate PAL provides powerful benefits for the overall health and well-being of adolescents, surpassing the effectiveness of any medical treatment or drug [9,17].

Another significant predictor of health status is physical fitness (PF), which is most commonly determined by physical capacities including cardiorespiratory fitness, muscular strength, power, and flexibility, along with anthropometric indices. PF is related to less abdominal obesity, decreased incidence of cardiovascular diseases, and improved mental and skeletal health [18]. PF is also strongly linked to metabolic risk in children [19]. Additionally, one of the determinants of PF, cardiorespiratory fitness (CRF), is inversely correlated with morbidity and mortality, making it one of the most important determinants of health status [20]. A longitudinal study reported that changes in CRF over 8 years can be a significant predictor of changes in body fat in adolescents (i.e., improved CRF leads to lower body fat percentage), which additionally highlights the importance of CRF as an indicator of overall health [21]. Collectively, PF is considered to be an important factor influencing improved health among adolescents, who are in a sensitive period of development, which means it is crucial to investigate and make improvements accordingly. Meanwhile, studies have confirmed negative changes in PF in children and adolescents during the COVID-19 pandemic, considering such trends to be a direct consequence of decreased PALs [22,23].

Significant positive associations between PAL and PF are well known [24]. For example, a significant linear relationship was found between PAL and CRF among US adolescents [25]. Similarly, a positive association between meeting physical activity guidelines and having aerobic and muscular fitness was found among Asian adolescents [26]. Moreover, a review of school-based interventions for increasing PAL showed that children and adolescents who increased their PAL had improved cardiometabolic health, as expressed by measures of CRF [27]. Last, but not least, body mass index, as an indicator of metabolic health, was inversely correlated with PAL among US adolescents [28]. Altogether, these data highlight the clear positive relationship between PAL, PF, and health status in young people.

From the overview of previous literature, it is clear that the PAL of children and adolescents decreased as a result of the COVID-19 pandemic and imposed lockdown measures, and that such changes also negatively influenced their PF, which was also the case among Croatian adolescents [3,5,23]. Additionally, investigating PAL and its association with PF is important in order to elucidate how to improve the health of adolescents [27,29]. However, to the best of our knowledge, there is a lack of studies examining this issue during the COVID-19 pandemic, and no study has examined this problem in southeastern Europe. Knowing about a problem can be of particular importance, especially in adolescents, considering: (i) the systematic evidence of negative trends in PAL and PF in this age group even under regular circumstances, and (ii) the additional detrimental effects of the COVID-19 pandemic on both PAL and PF. Additionally, previous research has shown that the PAL and PF indices differ between boys and girls during both regular and crisis situations, with boys having higher PAL and better PF than girls and a greater decrease in PAL during the COVID-19 pandemic than girls [5,30]. Based on the previous literature, we hypothesized that PAL is positively associated with PF irrespective of gender and the period during which the study was carried out. The aim of this research was to investigate the gender-specific associations between objectively measured PAL and PF indices (CRF, muscle strength, power, flexibility, and anthropometric indices) among Croatian high school adolescents during the COVID-19 pandemic.

## 2. Materials and Methods

### 2.1. Participants and Study Design

This cross-sectional study included 150 adolescents (101 females, 49 males) aged 14–18 years from Croatia. They were all attending high school. All students included in this study were in good health and did not have any injury or illness that might have prevented them from participating in PF tests. Participants (or legal guardians of minor students) signed informed consent before the study was initiated. The informed consent included a question asking whether students had any illness or injury. Students who reported any medical condition were excluded from the study. The study was approved by the Ethical Board Faculty of the Kinesiology University of Zagreb, Croatia (Ref. no. 25/2021, dated 16 July 2021).

### 2.2. Variables and Measurements

This cross-sectional study was carried out late in 2021, during the period of the declared COVID-19 pandemic. Testing was conducted at the start of the 2021–2022 school year, when there were no strict lockdown measures but some preventive strategies had been recommended (e.g., 2 m social distancing, travel restrictions). Variables included anthropometric indices, PF tests, and objectively assessed PAL. Although the measurement of participants was carried out over a 2 month period (due to a limited number of pedometers), each participant was measured on all variables within 10 days. At the time when the measurements were carried out, students were participating in the regular school program; however, during the previous school year and one month after testing, the Croatian government declared lockdown, and the Ministry of Science and Education requested that schools switch to online teaching (i.e., e-learning). All tests used are part of the standard fitness testing battery used in Croatian high schools, and the assessments were performed by experienced sport scientists and physical education teachers.

Anthropometric indices included body mass, body height, and calculated body mass index (BMI = body mass (kg)/body height (m)^2^).

PF was evaluated using tests examining CRF, power capacity, abdominal strength, and flexibility. The tests are part of the standard fitness evaluation in Croatian high schools and have been proven valid and reliable [31].

The beep test (also known as the pacer test or multilevel shuttle run test) was used to evaluate CRF. Students had to run 15 m from one line to another. This is a variation of the standard 20 m test, as a shorter distance was considered to be more appropriate for children and adolescents [32]. They had to follow the pace directed by a pre-recorded sound (beep). The pace increased at each level, and the test ended when students were not able to follow the pace. Test achievement was judged according to the number of 15 m runs performed within the time intervals determined by the beeps.

The broad jump (i.e., standing long jump) was used to evaluate power and was tested on a standardized jumping mat (Ghia Sport, Pazin, Croatia). Students performed a maximum forward jump using the arm swing. They made 3 jumps, and the longest jump (in cm) was considered for further analysis [33].

The sit-and-reach test was used to assess flexibility and was performed on a standardized wooden box. Participants placed the soles of their feet against the box, in the sitting position. They had to bend forward and reach as far as possible, and hold the position for at least 3 s. The best score out of 3 trials (in cm) was further analyzed [34].

Sit-ups, a measure of abdominal strength, were performed for 30 s. Students had to lift their torso while lying on their back, with bent knees and palms placed on their thighs. They had to pass the kneecaps with their hands. The result was the number of sit-ups in 30 s during only one trial [35].

PAL was measured using a commercially available Yamax SW200 pedometer (Yamasa Tokei Keiki Co. Ltd., Tokyo, Japan), which has been proven valid [36]. Students were asked to wear a pedometer for an entire week (including the weekend), and the result was calculated as the total number of steps.

### 2.3. Statistical Analysis

The Kolmogorov–Smirnov test was used to determine the normality of distribution of included variables (please see Appendix A for results), and since all variables met the normality assumption, parametric analyses were performed. Descriptive statistics included mean and standard deviation. The independent samples *t*-test was used to define gender differences for all variables.

The analysis of associations between PAL and PF was performed in several phases. First, Pearson’s correlation coefficient was calculated between PAL and PF variables for the total sample (i.e., without dividing boys and girls), and then stratified by gender. Since correlations showed significant differences in associations for boys and girls (see Section 3), the subsequent analyses were gender-stratified. In the next phase, Ward’s hierarchical clustering method based on Euclidean distance was used for multivariate taxonomic classification of participants according to their PF status (not including anthropometric indices). The formed groups/clusters were then identified using discriminant canonical analysis (DISCRA). Finally, the differences in PAL among the identified clusters were checked using one-way analysis of variance (ANOVA), with subsequent post hoc Scheffe test. The cluster analysis statistics allowed us to identify homogeneous groups of participants in the multivariate context (grouping was carried out multivariately based on PF status), which proved to be a valuable addition to the correlation analyses in previous studies [37,38,39].

A *p*-value of 0.05 was set for all procedures and Statistica v 13.5 (Tibco, Palo Alto, CA, USA) was used for all analyses.

## 3. Results

Boys were taller, heavier, had higher BMI, and achieved better results in the broad jump (power capacity), sit-ups (abdominal strength), and beep test (CRF). Girls achieved better results in the sit-and-reach test (flexibility). No significant differences in PAL were found between genders (Table 1).

Table 2 displays the correlation coefficients between studied variables, for the total sample and by gender. Apart from the logical correlations between anthropometric indices (e.g., body mass was strongly correlated with BMI) and between certain PF variables (e.g., a positive correlation was found between abdominal strength and power capacity), for the purpose of this study it is more important to recognize the generally poor relationships between anthropometrics, PF, and PAL. For example, none of the anthropometric indices was significantly correlated with PAL in the total sample or in the gender-stratified analysis (less than 1% common variance). Observing the total sample of participants, the only significant correlation between PF status and PAL was found for CRF (e.g., beep test) and PAL (6% common variance). However, gender-stratified correlation showed that even this low significant correlation was a consequence of the significant correlation between PAL and CRF among girls (8.4% common variance).

Figure 1 presents the clustering of participants based on their PF status. Three clusters were formed among girls (Figure 1A). Due to the smaller number of boys included in the study (n = 49), two clusters were identified among boys (Figure 1B).

The results of discriminant canonical analysis with previously identified PF-based clusters as the grouping variable showed that girls in cluster 3 had superior PF, particularly in CRF and abdominal strength (root 1). Meanwhile, girls in cluster 1 had better flexibility than their peers in cluster 2 (Table 3).

Discriminant analysis revealed differences between homogeneous groups of boys in CRF, power capacity, and abdominal strength, with boys in cluster 1 having better PF than their peers in cluster 2 (Table 4).

Descriptive statistics of the clusters in PAL and differences calculated by ANOVA are presented in Figure 2. ANOVA revealed significant differences (F-test: 4.11, *p* < 0.05) among girls, with cluster 3 having the highest PAL (significant post hoc differences between cluster 3 and cluster 1) (Figure 2A). No significant differences in PAL (F-test = 0.98, *p* > 0.05) were found among boys when PF-based clusters were compared (Figure 2B).

## 4. Discussion

This research has several important findings: (1) CRF and PAL were positively correlated in the total sample and among girls, while there was no significant correlation among boys; (2) correlations between PAL and broad jump, sit-ups, and sit-and-reach tests were not significant; (3) the associations between PAL and anthropometric indices were not significant. Therefore, our initial study hypothesis can only be partially accepted.

### 4.1. Physical Activity and Physical Fitness

A strong correlation between PF and PAL was not found, and CRF was correlated with PAL only among girls. Meanwhile, the results of previous studies conducted worldwide showed consistent positive correlations between PAL and CRF. For example, PAL was positively associated with CRF among Portuguese youth evaluated by the Progressive Aerobic Cardiovascular Endurance Run test [40]. In another study, children who improved their PAL also had improved CRF [21].

In explaining the differences between our results and those of previous studies, we must note that the cited studies used accelerometers and we used pedometers, both of which are used to assess PA, but in different ways and with different outputs. Specifically, an accelerometer records total PA and time spent doing activities at different intensities, while a pedometer indicates total activity expressed only by step count [41]. Simply put, a pedometer is not able to determine the actual intensity of physical activity (i.e., light, moderate, or vigorous). This is important, as it was previously emphasized that moderate-to-vigorous physical activity has better effects on CRF than light physical activity [40]. Thus, this could also be the reason why we did not record an association between PAL and CRF among boys. Generally, boys participate in more vigorous-intensity sports than girls, and a pedometer is simply unable to indicate the disparities in intensity leading to the result that the boys and girls in our study had similar PALs. Additionally, in previous studies, girls were found to be less physically active both during and after school, which might lead to detrimental effects on CRF [42]. If we consider that the motor capacities evaluated in this study are highly affected by exercise at the proper intensity (i.e., moderate or vigorous), the lack of associations found between PAL and motor capacity could be explained similarly (i.e., pedometers have limited applicability in high-intensity sports such as handball and football).

Additionally, our study did not confirm the strong association usually found between PAL and anthropometric indices among high school students. There is no doubt that this is related to the fact that, besides body mass index, we did not include other measures of body composition [43]. Indeed, it has been reported that body fat percentage is the main factor that drives the interaction between PAL and CRF [44]. Additionally, lower body fat percentage has been associated with higher levels of vigorous physical activity [45]. Meanwhile, BMI is frequently questioned as a measure of body composition simply because high values can be a result of increased muscle mass, which is particularly likely in athletes [46]. Another possible explanation may come from a previous part of the discussion; PAL was measured by pedometers, which are not applicable as a measurement tool for some forms of physical exercise (team sports, combat sports, etc.). On the other hand, it is known that contact sports involve physical activity with the highest energy expenditure, and consequently have the most impact on changes in body composition indices [47].

### 4.2. Possible Mediating Effects of the COVID-19 Pandemic on the Association between Physical Activity Levels and Physical Fitness

As noted in the Introduction, the COVID-19 pandemic dramatically influenced everyday life, and changes were especially evident in children and adolescents [3,48]. In brief, as a result of the lack of organized sports and regular physical education, the PAL of children and adolescents significantly decreased, which consequently negatively influenced their fitness status [49]. It seems that this negative trend influenced even the association between PAL and PF, resulting in a relatively low correlation between these variables in our study. Because of the lack of empirical data (as mentioned previously, there is an evident lack of studies on this problem during the COVID-19 pandemic), we cannot discuss this issue with certainty, but we will attempt to provide possible explanations.

The first explanation for the relatively low correlation between PAL and PF is simply that it is a mathematical/statistical issue. In brief, all types of correlational analyses are influenced by the variance of the results, since the correlation is proportional to the covariance of the variables. Mathematically, given two variables *A* and *B*, their correlation is defined as follows:*Correlation* = *Covariance* (*A*, *B*)/(*Standard Deviation* (*A*) × *Standard Deviation* (*B*))

The divisor in the equation has a scaling effect on the covariance so that the resulting correlation will lie between −1 and +1. So, all other things being equal, reducing the covariance will reduce the correlation [50]. In our case, we can speak about a decrease in the covariance simply because of the reduced variance of PAL and PF due to the COVID-19 pandemic. Indeed, studies have repeatedly confirmed that, along with a significant decrease in the average results of PAL during the pandemic period, the variance of PAL also decreased, and the same trends were found with PF [8,51]. One could argue that the correlation and reduced variance of the variables would not influence the results of our second set of statistics (e.g., cluster analysis → discriminant analysis → ANOVA). However, even in these analyses, the variance of results is a significant factor evidencing possible associations between variables. Specifically, in this set of analyses, the differentiation between/among clusters is highly dependent on the variance of the results, resulting in relatively small differences between clusters and consequently a limited possibility to identify the studied associations.

The second explanation for the relatively poor correlation between PAL and PF is more contextual. As stated in the Introduction, COVID-19 influenced adolescents’ behaviors in numerous ways. Most importantly, PAL decreased (due to a lack of sports, home schooling, and limited free play) [52]. In the meantime, screen time and overall sedentarism increased [53]. However, changes in PAL and PF during the pandemic did not follow the same pattern as in regular circumstances, during which they change simultaneously in all population subgroups. Additionally, and specific to the pandemic period, it was found that certain factors were associated with PAL among children and adolescents. In some cases, these influencing factors were relatively “non-standard”, such as familial conflict (with better PAL and smaller decrease in PAL during the pandemic among adolescents who reported less conflict with their family members) [7].

It is also important to note that the decrease in PAL during the pandemic was not equal for all subpopulations, but was most prevalent among adolescents who were more physically active in the pre-pandemic period (e.g., urban adolescents and boys) [6,49]. As a result, PF changed most dramatically among those adolescents who were the most fit and (logically) were the most physically active in the pre-pandemic period (e.g., athletes). Of course, this does not mean that athletes suddenly became unfit compared to their non-athletic peers; it simply means that the differences in PF between athletic and non-athletic adolescents during the pandemic were not as evident, as was the case in the pre-pandemic period.

As a support to what was previously stated about “non-standard” changes in PAL and PF during the pandemic, we can highlight the lack of difference in PAL between boys and girls in our study (see Table 1 for details), which was also relatively unexpected, since previous studies regularly reported that boys were more physically active than girls [30]. Finally, our results highlight the lack of a significant correlation between PAL and PF, specifically in boys, which additionally supports the idea that the influence of the COVID-19 pandemic on PAL and PF was not systematic. Therefore, the fact that PAL and PF changed differentially during the pandemic than in normal life circumstances could be considered as a plausible explanation for certain mediating effects of the COVID-19 pandemic on the studied association between PAL and PF among high school adolescents.

### 4.3. Limitations and Strengths

The main limitation of this study is the cross-sectional nature of the investigation. Thus, our results should be interpreted with caution, since we are not able to speak with certainty about the cause-and-effect relationship between variables. In other words, although it is generally considered that PAL actually influences PF (i.e., higher PAL “results” in better PF), the opposite causality is also possible (i.e., children with better PF are more physically active simply because they feel more confident in physically demanding activities). Additionally, we did not measure participants before the COVID-19 pandemic, so we cannot discuss its mediating effects on the association between PAL and PF with certainty. Finally, although we objectively measured PAL using pedometers, this measuring equipment has certain limitations in evaluating physical activity during high-intensity activities (e.g., contact sports such as soccer, basketball, and handball). Since we did not specifically examine the types of sports students were involved in outside the school setting, we could not discuss this issue in more depth.

This is a rare study topic globally, and this is probably the first study in the region (Southern Europe) to report associations between objectively measured PAL and PF during the COVID-19 pandemic. Additionally, the testing was carried out by the same evaluators in a standardized setting, which are important strengths of the investigation. Knowing that physical activity is one of the pillars of lifestyle medicine, we hope that this study will initiate further investigations on the topic.

## 5. Conclusions

The results of this study show that the association between PAL and PF in adolescents during the COVID-19 pandemic was generally low. While there are some methodological problems that could have influenced the results of the correlational analyses (e.g., the measurement tool used to evaluate PAL (pedometer) is not applicable for some high-intensity activities, such as contact sports), our findings could theoretically be at least partially attributed to the COVID-19 pandemic and the changes in PAL and PF that occurred as a consequence of various measures imposed in this period (i.e., lockdowns and social distancing).

While the associations between PAL and specific measures of PF may indicate important public health problems, especially among children and adolescents, it is important to note that CRF was positively associated with PAL in girls, even during the pandemic. This points to the specific positive correlation between overall physical activity and CRF as one of the most important health indicators. Therefore, future studies should focus on evaluating factors that could potentially be correlated with CRF in (late) adolescent boys. This is especially important considering that participation in sports, as the factor with the most influence on overall PF in children, significantly decreases in this period of life. Considering our findings showing a low association between PAL and PF in the pandemic period, this study could be used as a preliminary reference regarding the need to improve PAL and PF during pandemics or similar situations, but also during normal life periods, in order to prevent deterioration of adolescents’ healthy habits and health in general.

## Figures and Tables

**Figure 1 jfmk-08-00131-f001:**
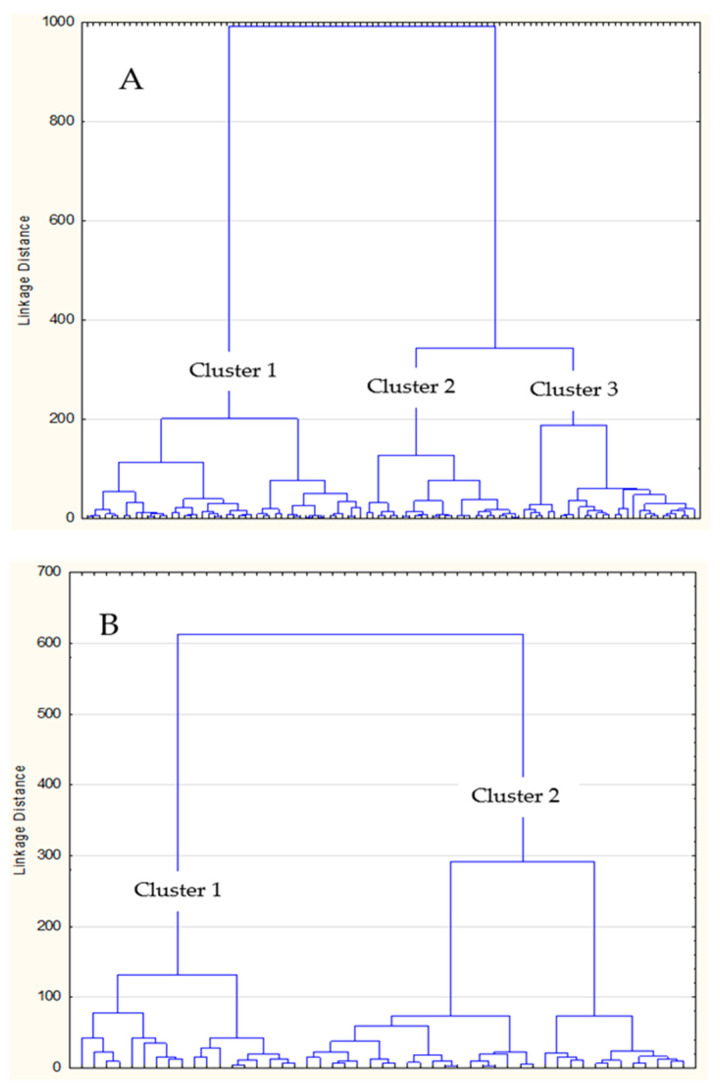
Results of multivariate clustering of (**A**) girls and (**B**) boys based on their physical fitness status.

**Figure 2 jfmk-08-00131-f002:**
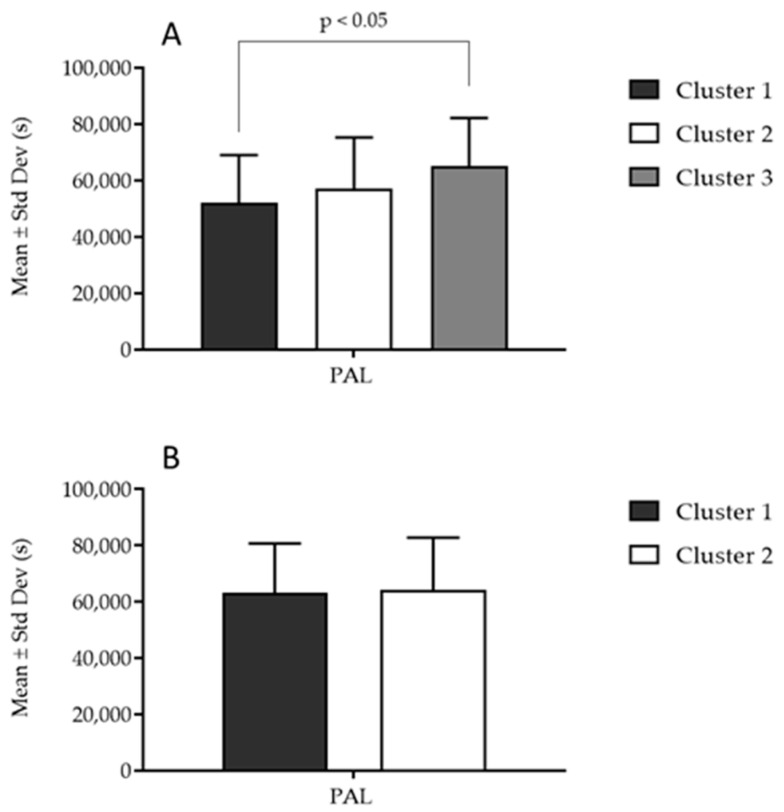
Differences between clusters based on physical activity levels among (**A**) girls and (**B**) boys.

**Table 1 jfmk-08-00131-t001:** Descriptive statistics and differences between genders.

	Boys (N = 49)		Girls (N = 101)		*T*-Test	
	Mean	SD	Mean	SD	*t*-Value	*p*-Value
Body height (cm)	180.84	7.64	168.24	6.67	10.37	0.001
Body mass (kg)	74.44	13.11	59.35	10.04	7.38	0.001
Body mass index	22.70	3.49	20.92	2.94	3.72	0.01
Broad jump (cm)	215.51	30.22	166.58	24.23	10.66	0.001
Sit-and-reach (cm)	6.90	9.59	13.31	6.98	4.65	0.001
Sit-ups (number)	65.98	10.98	52.20	11.64	6.92	0.001
Beep test (level)	11.77	2.74	9.62	9.50	6.91	0.001
Step count	61,828.63	24,196.94	59,804.20	19,384.67	0.55	0.58

**Table 2 jfmk-08-00131-t002:** Correlations between studied variables for total sample (n = 150), girls (n = 101), and boys (n = 49) (* significant correlation at *p* < 0.05).

		Body Height	Body Mass	Body Mass Index	Broad Jump	Sit-and-Reach	Sit-Ups	Beep Test
Body mass	Total sample	0.68 *						
	Girls	0.53 *						
	Boys	0.51 *						
Body mass index	Total sample	0.23 *	0.87 *					
	Girls	0.09	0.89 *					
	Boys	0.09	0.90 *					
Broad jump	Total sample	0.49 *	0.20 *	−0.07				
	Girls	0.02	−0.25 *	−0.29 *				
	Boys	0.23	−0.23	−0.38 *				
Sit-and-reach	Total sample	−0.28 *	−0.16 *	−0.02	−0.05			
	Girls	−0.08	0.11	0.18	0.22 *			
	Boys	−0.06	−0.05	−0.03	0.30 *			
Sit-ups	Total sample	0.39 *	0.25 *	0.07	0.61 *	0.06		
	Girls	0.12	−0.01	−0.08	0.40 *	0.25 *		
	Boys	0.07	−0.03	−0.05	0.49 *	0.39 *		
Beep test	Total sample	0.35 *	0.07	−0.15	0.67 *	−0.08	0.49 *	
	Girls	0.02	−0.23 *	−0.28 *	0.49 *	0.13	0.36 *	
	Boys	0.08	−0.32 *	−0.40 *	0.59 *	0.10	0.27	
PAL	Total sample	0.01	0.00	−0.01	0.02	0.06	0.07	0.25 *
	Girls	0.03	−0.01	−0.03	0.07	0.06	0.06	0.29 *
	Boys	−0.10	−0.05	−0.01	0.13	0.11	0.05	0.23

**Table 3 jfmk-08-00131-t003:** Results of discriminant canonical analysis of girls.

	Root 1	Root 2
Beep test	0.56	0.03
Broad jump	0.11	−0.02
Sit-ups	0.51	−0.09
Sit-and-reach	−0.04	0.41
Canonical R	0.41	0.32
Wilks’ lambda	0.65	0.74
*p*-value	0.01	0.03
Centroid: Cluster 1	−0.54	0.55
Centroid: Cluster 2	−0.11	−0.32
Centroid: Cluster 3	0.87	0.05

**Table 4 jfmk-08-00131-t004:** Results of discriminant canonical analysis of boys.

	Root 1
Beep test	0.41
Broad jump	0.32
Sit-ups	0.38
Sit-and-reach	−0.09
Canonical R	0.37
Wilks’ lambda	0.79
*p*-value	0.01
Centroid: Cluster 1	0.47
Centroid: Cluster 2	−0.40

## Data Availability

Data are available from the corresponding author upon reasonable request.

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
