# Peer review of "Mediating Effects of the COVID-19 Pandemic on the Associations between Physical Activity and Physical Fitness; Cross-Sectional Study among High School Adolescents"

_jfmk, 2023, doi:10.3390/jfmk8030131_

Round 1

Reviewer 1 Report

Abstract

Line 11: “The COVID-19 pandemic and imposed measures of social-distancing caused negative changes in physical activity levels (PAL) and physical fitness (PF) in children and adolescents” - The study is about adolescents. Reference about children doesn’t make sense.

Introduction

Line 57-69 – These data is about adolescents?

After I read this section, I can’t understand whether this study is essential or not, because the data about PAF, PF, CRF and body composition of Croatian adolescents is unspecified. The authors only wrote in line 47 that “Croatian adolescents, with only 23% of adolescents reaching the physical activity guidelines”.  The problem of the study isn’t totally justified.

Moreover, this study aimed: “Therefore, this research aimed to investigate the gender-specific associations between objectively measured PAL and indices of PF….” – Here, it is the first time in this section that appears gender. Thus, this strength the gap of justification of this study.

Methods

Who performed the variables’ assessment? 

The choice about these tests to evaluated PF should be justified.

Line 158 – remove extra .

The authors used the Kolmogorov-Smirnov test, but they didn’t write about their result and the impact of this test on statistical analysis.

Anova analysis is usually to compare means than 3 or more groups.

In this section, the authors identified the analysis about girls and boys….however this analysis in fully unjustified until this point.

The correlation analysis between gender and variables isn’t also justified.

Discussion / Conclusion

Please, remove the lines 225-228.

Line 241 pelase, fix “beforementionated”

Line 255 and 256 This affirmation should be supported by a reference.

In general, this discussion need more explores about covid 19 variable. We can not write only about COVID 19, but we need to make a discussion about consequences of COVID 19 (e.g., more screen time), and the relation with yours results.

Author Response

Abstract

Line 11: “The COVID-19 pandemic and imposed measures of social-distancing caused negative changes in physical activity levels (PAL) and physical fitness (PF) in children and adolescents” - The study is about adolescents. Reference about children doesn’t make sense.

RESPONSE: Thank you for noticing this. We removed “children”.

Introduction

Line 57-69 – These data is about adolescents?

RESPONSE: It begins with physical fitness and health in general and later focuses on physical fitness and its influence on health specifically in adolescents.

After I read this section, I can’t understand whether this study is essential or not, because the data about PAF, PF, CRF and body composition of Croatian adolescents is unspecified. The authors only wrote in line 47 that “Croatian adolescents, with only 23% of adolescents reaching the physical activity guidelines”.  The problem of the study isn’t totally justified.

RESPONSE: We mentioned the Croatian adolescents in this part because previous studies reported an alarmingly high number of insufficiently active Croatian adolescents, with girls being less active than boys. That is why aimed to investigate the PAL and PF specifically in Croatian adolescents-to elucidate the possible reasons of this problem, and with the emphasis on the COVID-19 “generation” as it is even more problematic due to restrictions of movements during the lockdowns. The explanation is now added in the Introduction, text reads: “From the previous literature overview, it is clear that the PAL of children and adolescents decreased as a result of the COVID-19 pandemic and imposed measures of lockdown, and that such changes in PAL negatively influenced even their PF, which also appeared among Croatian adolescents [20,26,27].”

Moreover, this study aimed: “Therefore, this research aimed to investigate the gender-specific associations between objectively measured PAL and indices of PF….” – Here, it is the first time in this section that appears gender. Thus, this strength the gap of justification of this study.

RESPONSE: Thank you for noticing. We added a justification why we gender-stratified the  investigation in this research. Text now reads: “Also, previous research elucidated that boys and girls differ in PAL and PF indices during both regular and crisis situations; with boys having higher PAL and better PF than girls and boys decreasing their PAL during the COVID-19 pandemic more than girls [28,29].” Please see the last paragraph of the Introduction.

Methods

Who performed the variables’ assessment?

RESPONSE: Thank you for noticing that we forgot to mention this important information. It is now added in the Variables and measurements section. Text reads: “The assessments were performed by experienced sport scientists and physical education teacher”.

The choice about these tests to evaluated PF should be justified.

RESPONSE: Thank you for noticing it. The test we included tests are part of standard fitness test battery in Croatian high schools and have been proven valid and reliable previously. Text reads: “All tests used are part of the standard fitness testing battery used in Croatian high schools, and the assessments were performed by experienced sport scientists and physical education teachers “ Please see lines 124-126

Line 158 – remove extra .

RESPONSE: It is now removed.

The authors used the Kolmogorov-Smirnov test, but they didn’t write about their result and the impact of this test on statistical analysis.

RESPONSE: The Kolmogorov-Smirnov test was used to determine the normality of the distribution of included variables, which means that it was used only for being able to choose correct statistical analysis based on whether the variables were normally (parametric tests) or non-normally (non-parametric tests) distributed. It is now explained in the text: “The Kolmogorov–Smirnov test was used to determine the normality of distribution of included variables, and since all variables met the normality assumption, parametric analyses were performed.”. (please see highlighted text; beginning of the Statistics subsection).

Anova analysis is usually to compare means than 3 or more groups.

RESPONSE: Yes, thank you for this comment. We changed the analysis into T-test. Table one is now changed accordingly. Also, we must apologize for some mistakes in the previous table 1 (some significant differences between genders were wrongly presented as non-significant). It is now corrected and text reads: “Boys were taller, heavier, had higher BMI, and achieved better results in the broad jump (power capacity), sit-ups (abdominal strength), and beep test (CRF). Girls achieved better results in the sit-and-reach test (flexibility). No significant differences in PAL were found between genders (Table 1).” (please see highlighted text at the beginning of the Results section)

In this section, the authors identified the analysis about girls and boys….however this analysis in fully unjustified until this point.

RESPONSE: As we have done a lot of studies about the physical activity and fitness of adolescents, we are aware that physical activity and fitness regularly differs between genders. That is why we have chosen to do the gender-stratified analysis, as we believed it will provide more detailed and accurate results. It is now clarified throughout the Introduction. Text reads: “Moreover, other factors have been shown to have an influence on changes in PAL during the pandemic [4]. Precisely, age (larger PAL decline was observed in younger than in older adolescents) [5], gender (boys had higher decrease in PAL than girls) [6], living environment (urban adolescents had higher decrease in PAL than rural adolescents) [7], and familial/parental factors (adolescents who conflicted with their parents had lower PAL) [8] have been shown to impact the changes in PAL during the pandemic”.

and

“The same trend is present among Croatian adolescents, with only 23% of adolescents (21% of girls, 27% of boys) reaching the physical activity guidelines standards [16]”.

and

“Also, previous research elucidated that boys and girls differ in PAL and PF indices during both regular and crisis situations; with boys having higher PAL and better PF than girls and boys decreasing their PAL during the COVID-19 pandemic more than girls [6,31]”.

The correlation analysis between gender and variables isn’t also justified.

RESPONSE: Please see previous comments about the gender-stratified analysis.

Discussion / Conclusion

Please, remove the lines 225-228.

RESPONSE: It is now removed; we apologise for this error.

Line 241 pelase, fix “beforementionated”

RESPONSE: It is now corrected.

Line 255 and 256 This affirmation should be supported by a reference.

RESPONSE: Thank you for suggesting this, it is now added. (Prentice, A.M.; Jebb, S.A. Beyond body mass index. Obesity reviews 2001, 2, 141-147.)

In general, this discussion need more explores about covid 19 variable. We can not write only about COVID 19, but we need to make a discussion about consequences of COVID 19 (e.g., more screen time), and the relation with yours results.

RESPONSE: Thank you for your suggestion. Accordingly, the discussion is systematically rewritten in some parts and we tried to highlight the consequences of COVID-19 pandemic. The changed text is evident in the last three paragraphs of the subsection 4.2 (please see text highlighted in yellow). Thank you!

Staying at your disposal!

Authors

Reviewer 2 Report

Overall, the design of the study is average and not very original. In addition, there are some issues that need to be addressed:

- The main issue is that the authors cannot state that the Mediating Effects of the COVID-19 are the responsable or have any relationship with PAL and/or PF. You have not measured participants before COVID-19. Change it or delete it in the title and in the text

Intro

- First and second paragraph can be combined. Otherwise the intro is too long and repetitive

- Include the hypothesis of your study before the aims

Methods

- How did you know the following? "All students that were in-102 cluded in this study were in good health...."

- Include references of the procedures followed in the beep test, sit-up test and sit and reach test

Results

- Table 1 analysis makes no sense. Morphology in boys and girls are significantly different. Delete it

- Why in boys ang girls the number of clusters are different? Is there any explanation?

Discussion

- Delete the 1st paragraph

- Why do you think you found that the associations between PAL and PF during the COVID-19 pandemic were generally low?

- Include in the limitations sections the first paragraph of the conclusions ("problems")

- It is too daring to state that the low relationships beetween PAL and PF  occurred as a consequence of various measures imposed in this period (i.e. lock-down and socialdistancing). You did not measure the PAL or PF of the participants before the COVID-19

English

A native speaker or a specialized company should review the whole article. It does not have meaningful mistakes but there are some minor spelling mistakes and expressions with scientific vocabulary missing. Some examples:

- Line 45. "worrying numbers;" Use scientific vocabulary

- Line 48. "Guidelines" Use "standards"

- Line 121. "Used tests are....." Use another word

- Line 235. Avoid personal forms such as "we"

Author Response

Overall, the design of the study is average and not very original. In addition, there are some issues that need to be addressed:

- The main issue is that the authors cannot state that the Mediating Effects of the COVID-19 are the responsable or have any relationship with PAL and/or PF. You have not measured participants before COVID-19. Change it or delete it in the title and in the text

RESPONSE: Thank you for this valuable suggestion. We agree with your statement and, thus, we have changed the title into: “Analysis of the Associations between Physical Activity and Physical Fitness among High School Adolescents during COVID-19 pandemic period”. Also, we added this issue in the Limitations section, text reads: “Also, we have not measured participants before the COVID-19 pandemic so we cannot undoubtedly discuss the mediating effects of the COVID-19 pandemic on the associations between PAL and PF”.

Intro

- First and second paragraph can be combined. Otherwise the intro is too long and repetitive

RESPONSE: We combined the first two paragraphs into one.

- Include the hypothesis of your study before the aims

RESPONSE: The hypothesis is now included before the aims, thank you.

Methods

- How did you know the following? "All students that were in-102 cluded in this study were in good health...."

RESPONSE: Thank you for this suggestion, we did not specify that in the text. We knew that students were in good health because all students had to sign informed consent (or their parents/legal guardians), in which was also included the question whether they have any illness or injury. The students which reported any medical condition were excluded from the study. This is also now added in the Methods section, Participants and study design (please see text highlighted in yellow). Text reads: “All students included in this study were in good health and did not have any injury or illness that might have prevented them from participating in PF tests. Participants (or legal guardians of minor students) signed informed consent before the study was ini-tiated. The informed consent included a question asking whether students had any illness or injury. Students who reported any medical condition were excluded from the study.”

- Include references of the procedures followed in the beep test, sit-up test and sit and reach test

RESPONSE: The references are now added; specifically:

Mraković, M.; Findak, V.; Metikoš, D.; Neljak, B. Developmental characteristics of motor and functional abilities in primary and secondary school pupils. Kinesiology 1996, 28, 57-65.

McClain, J.J.; Welk, G.J.; Ihmels, M.; Schaben, J. Comparison of Two Versions of the PACER Aerobic Fitness Test. Journal of Physical Activity and Health 2006, 3, S47-S57, doi:10.1123/jpah.3.s2.s47.

Thomas, E.; Petrigna, L.; Tabacchi, G.; Teixeira, E.; Pajaujiene, S.; Sturm, D.J.; Sahin, F.N.; Gómez-López, M.; Pausic, J.; Paoli, A.; et al. Percentile values of the standing broad jump in children and adolescents aged 6-18 years old. Eur J Transl Myol 2020, 30, 9050, doi:10.4081/ejtm.2019.9050.

Miyamoto, N.; Hirata, K.; Kimura, N.; Miyamoto-Mikami, E. Contributions of Hamstring Stiffness to Straight-Leg-Raise and Sit-and-Reach Test Scores. Int J Sports Med 2018, 39, 110-114, doi:10.1055/s-0043-117411.

Chandana, A.; Xubo, W. The test battery: Evaluate muscular strength and endurance of the abdominals and hip-flexor muscles. 2021

Results

- Table 1 analysis makes no sense. Morphology in boys and girls are significantly different. Delete it

RESPONSE: Thank you for noticing it. Definitively, the results in table were not correct, and we sincerely apologize for that. It is corrected now, and as evident, boys and girls differ in practically all variables (but PAL). However, we must mention that following the comments of the 1st and 3rd reviewer in this version of the manuscript we calculated t-test (and not ANOVA) to determine the differences between genders. Text related to Table 1 now reads: “Boys were taller, heavier, had higher BMI, and achieved better results in the broad jump (power capacity), sit-ups (abdominal strength), and beep test (CRF). Girls achieved better results in the sit-and-reach test (flexibility). No significant differences in PAL were found between genders (Table 1).” (please see highlighted text at the beginning of the Results subsection). Also, corrected results in Table 1 are noteb by highlighted text.

- Why in boys ang girls the number of clusters are different? Is there any explanation?

RESPONSE: The different number of clusters in boys and girls is mostly a consequence of number of participants per gender (101 girls and 49 boys), and the fact that homogenisation of the participation was somehow specific (please see Figure 1B). As it is evident in Figure 1B forming three clusters for boys would be inappropriate, but we should form four clusters – two subclasters of Cluster 1 – on the left side of the graph; and two subclusters of Cluster 2 – right side of the graph). Hence, it will result in only 10-12 participants in each cluster which will make further statistical analyses underpowered. It is briefly explained in the results section now, and text reads: “Figure 1 presents the clustering of participants based on their PF status. Three clusters were formed among girls (Figure 1A). Due to the smaller number of boys in-cluded in the study (n = 49), two clusters were identified among boys (Figure 1B).” (please see text before Figure 1), thank you.

Discussion

- Delete the 1st paragraph

RESPONSE: It is now deleted, thank you for noticing.

- Why do you think you found that the associations between PAL and PF during the COVID-19 pandemic were generally low?

RESPONSE: Thank you for your question. Indeed, the similar suggestion was brought by 1st Reviewer, and this issue is now discussed more profoundly in the subheading 4.2. (please see text highlighted in yellow).

- Include in the limitations sections the first paragraph of the conclusions ("problems")

RESPONSE: It is now included in the limitations. Text reads: “Although we objectively measured the PAL by pedometers, this measuring equipment has certain limitations in evaluating the physical activity during some high-intensity activities (i.e., contact sports such as soccer, basketball, handball)”.

- It is too daring to state that the low relationships beetween PAL and PF  occurred as a consequence of various measures imposed in this period (i.e. lock-down and socialdistancing). You did not measure the PAL or PF of the participants before the COVID-19

RESPONSE Thank you for this suggestion. Indeed, we are unable to undoubtedly state it. Thus, we changed the text with intention to present this notion as theoretical and not a fact. Text now reads: “While there are some methodological problems that could influence the results of the correlational analyses (i.e., measurement tool used for the evaluation of the PAL (pedometers) are not applicable during some high-intensity activities, such as contact sport), our findings could theoretically be at least partially a consequence of the COVID-19 pandemic and changes in PAL and PF that occurred as a consequence of various measures imposed in this period (i.e., lock-down and social-distancing).” Also, the title of the article is changed accordingly, and now reads: “Analysis of the Associations between Physical Activity and Physical Fitness among High School Adolescents during COVID-19 pandemic period”

Comments on the Quality of English Language

English

A native speaker or a specialized company should review the whole article. It does not have meaningful mistakes but there are some minor spelling mistakes and expressions with scientific vocabulary missing. Some examples:

- Line 45. "worrying numbers;" Use scientific vocabulary

RESPONSE: Thank you; changed into “alarming figures”.

- Line 48. "Guidelines" Use "standards"

RESPONSE: Amended accordingly.

- Line 121. "Used tests are....." Use another word

RESPONSE: we changed it into elements. Text reads: “Used tests are elements of standard fitness evaluation in Croatian high schools and have been proven valid and reliable previously”.

- Line 235. Avoid personal forms such as "we"

RESPONSE: Amended accordingly. Text now reads: “Strong correlation between PF and PAL was not found, and CRF was correlated with PAL only in girls”.

Following all previous corrections, the revised version of the manuscript is checked for language quality by language editing service (details are provided to Editor).

Staying at your disposal!

Authors

Reviewer 3 Report

The manuscript under reviewing addresses a presumably mediating effect of covid-19 pandemic on the physcial activity levels and, consequently, on the physical fitness status of a high school students sample. This goal is achieved, partially in my perspective, through a set of statistics models presented in a chain way, which reveals the Authors seem to know from beggining how to work on their data.  However, it seems much more attention was given to data presentation than to its content interpretation, which I further detail below.

(lines 33-36): It is not just the lack of organized sports, recreational activities and physical education classes that explain the decrease in physical activity rates during the pandemic, but other factors have also been shown to have an influence, such as age, gender and socioeconomic status. Authors should approach this topic in a holistic way.

Reading suggestion:

Rossi, L., Behme, N., & Breuer, C. (2021). Physical activity of children and adolescents during the COVID-19 pandemic—A scoping review. International journal of environmental research and public health, 18(21), 11440. 

There sould have uniformity in the characterization of the measurement units that each variable/test provides, as, for example, was done in the case of sit-ups (number of accomplishments in 30 seconds) and PAL (total number of steps taken), which did not occur for the adapted shuttle-run test (number of 15m runs performed within the time intervals determined by the beeps) and for the broad-jump (greater distance in centimeters).

(lines 113-117): In what exactly period of the school year did the measurement procedures take place? For how long have the students participated in the regular school program, before the lockdown declared by th croatians authorities?  Maybe the mediating effects the Authors propose to study is not reachable through the undertaken experimental design, insofar, accordingly what is argued in the introduction (lines 33-36 and lines 54-56), those effects were more  evident throughout lockdown periods, which seems have not occured during the data collection.

(lines 149-150): if the intention is to define gender differences,  would have not been more appropriated to run a T-teste, rather than a ANOVA?

Figure 1: In order to provide a better illustrative visual representation upon the clusters' formation, both dendograms (1A and 1B) should not be alongside from one another, but rather above and below, allowing to optmize its height and width and thus becoming easier to readers compreehend the clustering roots.

(lines 235-236): Why there may exist a correlation between CRF and PAL in girls? What may help to understand this finding? The Authors should address this issue more thoroughly, insofar such differences found could be explained by a girls´ tend to be less physically active than boys both during and outside the school day (Nyberg, Nordenfelt, Ekelund, & Marcus, 2009; Long et al., 2013; Calvert, Mahar, Flay & Turner, 2018), which potencially might have a detrimental effect on her cardiorespiratory capacity development.

Reading suggestion:

Nyberg, G. A., Nordenfelt, A. M., Ekelund, U., & Marcus, C. (2009). Physical activity patterns measured by accelerometry in 6-to 10-yr-old children. Medicine and science in sports and exercise, 41(10), 1842-1848. 

Long, M. W., Sobol, A. M., Cradock, A. L., Subramanian, S., Blendon, R. J., & Gortmaker, S. L. (2013). School-day and overall physical activity among youth. American journal of preventive medicine, 45(2), 150-157. 

Calvert, H. G., Mahar, M. T., Flay, B., & Turner, L. (2018). Classroom-based physical activity: minimizing disparities in school-day physical activity among elementary school students. Journal of Physical Activity and Health, 15(3), 161-168. 

(lines 287-300):  it is curious how the Authors justify the low correlation found "blaming"  the covariance, which is an issue there is no control of in advance, when other questions were worth to be included into the discussion, such as, for example:

1) What was the purpose of undergo data to a hierarchical cluster analysis? 

2) Are there other studies that have follow the same or similar statiscal procedure?

3)  Are there findings in literature that help to understand why 3 clusters were formed for girls and 2 for boys? Regardless of yes or no, what is the Authors' opinion towards this finding?

Reading suggestions:

López-Gil, J. F., Brazo-Sayavera, J., García-Hermoso, A., Camargo, E. M. d., & Yuste Lucas, J. L. (2020). Clustering patterns of physical fitness, physical activity, sedentary, and dietary behavior among school children. Childhood Obesity, 16(8), 564-570. 

Bachner, J., Sturm, D. J., García-Massó, X., Molina-García, J., & Demetriou, Y. (2020). Physical activity-related profiles of female sixth-graders regarding motivational psychosocial variables: a cluster analysis within the CReActivity Project. Frontiers in Psychology, 11, 580563. 

(lines 344-345): In the studied sample, were there students who practiced contact sports? The conclusions must be limited to the studied sample characteristics. Was there any data collection about this? 

Author Response

Reviewer 3

The manuscript under reviewing addresses a presumably mediating effect of covid-19 pandemic on the physcial activity levels and, consequently, on the physical fitness status of a high school students sample. This goal is achieved, partially in my perspective, through a set of statistics models presented in a chain way, which reveals the Authors seem to know from beggining how to work on their data.  However, it seems much more attention was given to data presentation than to its content interpretation, which I further detail below.

RESPONSE: Thank you for recognizing the importance of our investigation and providing us the opportunity to improve the quality of the manuscript.

(lines 33-36): It is not just the lack of organized sports, recreational activities and physical education classes that explain the decrease in physical activity rates during the pandemic, but other factors have also been shown to have an influence, such as age, gender and socioeconomic status. Authors should approach this topic in a holistic way.

RESPONSE: Thank you for noticing this. Other factors have been mentioned in the Discussion, but we did not state them in the Introduction. Thus, as we agree it is important, other factors classes that can explain the decrease in physical activity rates during the pandemic are now listed in the Introduction. Text now reads: “Moreover, other factors have been shown to have an influence on changes in PAL during the pandemic [4]. Precisely, age (larger PAL decline was observed in younger than in older adolescents) [5], gender (boys had higher decrease in PAL than girls) [6], living environment (urban adolescents had higher decrease in PAL than rural adolescents) [7], and familial/parental factors (adolescents who conflicted with their parents had lower PAL) [8] have been shown to impact the changes in PAL during the pandemic”.

Reading suggestion:

Rossi, L., Behme, N., & Breuer, C. (2021). Physical activity of children and adolescents during the COVID-19 pandemic—A scoping review. International journal of environmental research and public health, 18(21), 11440.

RESPONSE: Thank you! The suggested reference is now added in the text.

There sould have uniformity in the characterization of the measurement units that each variable/test provides, as, for example, was done in the case of sit-ups (number of accomplishments in 30 seconds) and PAL (total number of steps taken), which did not occur for the adapted shuttle-run test (number of 15m runs performed within the time intervals determined by the beeps) and for the broad-jump (greater distance in centimeters).

RESPONSE: Thank you for noticing it. The results on each test are now specified for each variable (please see highlighted text line 132-154).

(lines 113-117): In what exactly period of the school year did the measurement procedures take place? For how long have the students participated in the regular school program, before the lockdown declared by th croatians authorities?  Maybe the mediating effects the Authors propose to study is not reachable through the undertaken experimental design, insofar, accordingly what is argued in the introduction (lines 33-36 and lines 54-56), those effects were more  evident throughout lockdown periods, which seems have not occured during the data collection.

RESPONSE: Thank you for this comment. We conducted testing at the start of the school year 2021/2022, when there were no strict lockdown measures but some of the preventive strategies have still been recommended (e.g., 2-meter social distancing measure, traveling restrictions and similar). Also, as this period was shortly after the strict lock-down measures, people were still afraid to act “normally” in the everyday life (e.g., they were avoiding crowded places such as shopping malls and fitness centres). Thus, as we previously conducted studies during the pandemic, we wanted to check whether the pandemic period had long lasting effects on physical activity habits and consequently fitness status of adolescents. It is now added in the Participants an study design section. Text reads: “Testing was conducted at the start of the school year 2021/2022, when there were no strict lockdown measures but some of the preventive strategies have still been recom-mended (e.g., 2-meter social distancing measure, traveling restrictions)”.

(lines 149-150): if the intention is to define gender differences,  would have not been more appropriated to run a T-teste, rather than a ANOVA?

RESPONSE: Thank you for noticing this. For this version of the manuscript we calculated T-test and presented the results in the Table 1.

Figure 1: In order to provide a better illustrative visual representation upon the clusters' formation, both dendograms (1A and 1B) should not be alongside from one another, but rather above and below, allowing to optmize its height and width and thus becoming easier to readers compreehend the clustering roots.

RESPONSE: Thank you for your suggestion. The Figure 1 is amended accordingly.

(lines 235-236): Why there may exist a correlation between CRF and PAL in girls? What may help to understand this finding? The Authors should address this issue more thoroughly, insofar such differences found could be explained by a girls´ tend to be less physically active than boys both during and outside the school day (Nyberg, Nordenfelt, Ekelund, & Marcus, 2009; Long et al., 2013; Calvert, Mahar, Flay & Turner, 2018), which potencially might have a detrimental effect on her cardiorespiratory capacity development.

RESPONSE: Thank you for this valuable suggestion. This is now explained in more detail in the Manuscript. Text reads: “Precisely, boys participate in more vigorous-intensity sports compared to girls, and pedometers are simply unable to determine that disparities in the intensity which led to the result that boys and girls in our study had similar PAL. Also, in previous studies girls displayed being less physically active both during and after school time, which might lead to detrimental effects on CRF [40]”. Please see the “Physical activity and fitness” section of the Discussion.

Reading suggestion:

Nyberg, G. A., Nordenfelt, A. M., Ekelund, U., & Marcus, C. (2009). Physical activity patterns measured by accelerometry in 6-to 10-yr-old children. Medicine and science in sports and exercise, 41(10), 1842-1848.

Long, M. W., Sobol, A. M., Cradock, A. L., Subramanian, S., Blendon, R. J., & Gortmaker, S. L. (2013). School-day and overall physical activity among youth. American journal of preventive medicine, 45(2), 150-157.

Calvert, H. G., Mahar, M. T., Flay, B., & Turner, L. (2018). Classroom-based physical activity: minimizing disparities in school-day physical activity among elementary school students. Journal of Physical Activity and Health, 15(3), 161-168.

RESPONSE: Thank you! The reference is now included in the text accordingly: Nyberg, G.A.; Nordenfelt, A.M.; Ekelund, U.; Marcus, C. Physical activity patterns measured by accelerometry in 6- to 10-yr-old children. Med Sci Sports Exerc 2009, 41, 1842-1848, doi:10.1249/MSS.0b013e3181a48ee6.

(lines 287-300):  it is curious how the Authors justify the low correlation found "blaming"  the covariance, which is an issue there is no control of in advance, when other questions were worth to be included into the discussion, such as, for example:

1) What was the purpose of undergo data to a hierarchical cluster analysis?

2) Are there other studies that have follow the same or similar statiscal procedure?

3)  Are there findings in literature that help to understand why 3 clusters were formed for girls and 2 for boys? Regardless of yes or no, what is the Authors' opinion towards this finding?

RESPONSE: Thank you for your questions. Here are the responses:

  • Actually, the cluster analysis was proven to be valuable “alternative” (or “addition”) to correlation coefficients in some previous studies (see please later for details) because correlation coefficients are highly dependent on “averages”. Namely, if the scatterplot of correlated variables is dense in average results, the correlation will not likely be significant. On the other hand, hierarchical clustering allow to define “homogenous groups” of participants and (later) to identify the differences between diverse groups (“extremes”) which can not be done by correlation analysis. Also, clustering of the subjects was done on multivariate basis, which we considered as valuable alternative to univariate correlations.
  • Yes, indeed previous studies used such approach and are now added in the References
  • Geets-Kesic, M.; Maras, N.; Gilić, B. Analysis of the Association Between Health Literacy, Physical Literacy, and Scholastic Achievement; A Preliminary Cross-Sectional Study Among High-School Students From Southern Croatia. Montenegrin Journal of Sports Science & Medicine 2023, 12.
  • López-Gil, J.F.; Brazo-Sayavera, J.; García-Hermoso, A.; Camargo, E.M.; Yuste Lucas, J.L. Clustering Patterns of Physical Fitness, Physical Activity, Sedentary, and Dietary Behavior among School Children. Child Obes 2020, 16, 564-570, doi:10.1089/chi.2020.0185.
  • The different number of clusters in boys and girls is mostly a consequence of number of participants per gender (101 girls and 49 boys), and the fact that homogenisation of the participation was somehow specific (please see Figure 1B). As it is evident in Figure 1B forming three clusters for boys would be inappropriate, but we should form four clusters – two subclasters of Cluster 1 – on the left side of the graph; and two subclusters of Cluster 2 – right side of the graph). Hence, it will result in only 10-12 participants in each cluster which will make further statistical analyses underpowered. It is briefly explained in the results section now.

In general, the rationale for the statistical approach (cluster analysis) is now briefly explained in the Statistics subsection, and text reads: “The cluster analysis statistics allowed us to identify homogenous groups of partici-pants in the multivariate context (grouping was done multivariately on the basis of PF status), which was proven to be valuable addition to correlation analyses in previous similar studies”

Reading suggestions:

López-Gil, J. F., Brazo-Sayavera, J., García-Hermoso, A., Camargo, E. M. d., & Yuste Lucas, J. L. (2020). Clustering patterns of physical fitness, physical activity, sedentary, and dietary behavior among school children. Childhood Obesity, 16(8), 564-570.

Bachner, J., Sturm, D. J., García-Massó, X., Molina-García, J., & Demetriou, Y. (2020). Physical activity-related profiles of female sixth-graders regarding motivational psychosocial variables: a cluster analysis within the CReActivity Project. Frontiers in Psychology, 11, 580563.

RESPONSE: Thank you for this suggestion. Several references are now included regarding the clustering rationale in the Statistical analysis section:

  • Dalmaijer, E.S.; Nord, C.L.; Astle, D.E. Statistical power for cluster analysis. BMC Bioinformatics 2022, 23, 205, doi:10.1186/s12859-022-04675-1.
  • Geets-Kesic, M.; Savicevic, A.J.; Peric, M.; Gilic, B.; Zenic, N. Specificity of the Associations between Indices of Cardiovascular Health with Health Literacy and Physical Literacy; A Cross-Sectional Study in Older Adolescents. Medicina 2022, 58, 1316.
  • Geets-Kesic, M.; Maras, N.; Gilić, B. Analysis of the Association Between Health Literacy, Physical Literacy, and Scholastic Achievement; A Preliminary Cross-Sectional Study Among High-School Students From Southern Croatia. Montenegrin Journal of Sports Science & Medicine 2023, 12.
  • López-Gil, J.F.; Brazo-Sayavera, J.; García-Hermoso, A.; Camargo, E.M.; Yuste Lucas, J.L. Clustering Patterns of Physical Fitness, Physical Activity, Sedentary, and Dietary Behavior among School Children. Child Obes 2020, 16, 564-570, doi:10.1089/chi.2020.0185.

(lines 344-345): In the studied sample, were there students who practiced contact sports? The conclusions must be limited to the studied sample characteristics. Was there any data collection about this?

RESPONSE: Unfortunately, we didn’t collect the exact data on students’ practicing contact sports, but since first author of the study is a PE teacher in the school where study was done, she is very much aware that contact sports (i.e. soccer, handball, basketball) are very popular and almost certainly some of the children participated in it. This problem is now highlighted in the limitations, and text reads: “While we didn’t specifically examine the type of sport students were eventually involved at in out of school settings, we couldn’t discuss this issue more profoundly.” (please see Limitations and Strengths subsection at the end of Discussion).

Staying at your disposal!

Authors

Round 2

Reviewer 1 Report

Dear authors,

thank you for your work. 

However, I suggest minor reviews in this paper.

You need to notice the T-Test in Abstract section.

You did define why you used Kolmogor-Smirnov, but you didn't notice the results in the Results section. 

Regards.

Author Response

Dear Reviewer

Thank you for your additional comments, please see how we amended manuscript accordingly. 

You need to notice the T-Test in Abstract section.

RESPONSE: T-test is specified; please see highlighted text (in yellow)

You did define why you used Kolmogor-Smirnov, but you didn't notice the results in the Results section. 

RESPONSE: Results of the KS test are given in the supplementary table 1 and noted in the Statistics subsection (please see Supplementary table and text highlighted in yellow in subsection Statistics). Thank you!

Staying at your disposal!

Authors